# Boosting immune response against cervical cancer: A combined approach using oncolytic virus and targeted therapies

Hedieh Zargaran[1], Amir Ghaemi[2]*, Mohammad Shenagari[1,3]*, Mehdi Samadi[4]

1 Department of Microbiology, School of Medicine, Guilan University of Medical Sciences, Rasht, Iran,
2 Department of Influenza and Other Respiratory Viruses, Pasteur Institute of Iran, Tehran, Iran, 3 Cellular and Molecular Research Center, Faculty of Medicine, Guilan University of Medical Sciences, Rasht, Iran,
4 Department of Microbiology, Immunogenetics Research Center, Faculty of Medicine, Mazandaran University of Medical Sciences, Sari, Iran

* ghaem_amir@yahoo.com (AG); shenagari@gums.ac.ir (MS)

## Abstract

### Background

Cervical cancer remains a primary reason for cancer malignancy among women worldwide, primarily due to human papillomavirus (HPV) strains HPV16 and HPV18. Despite having access to vaccines, there are few treatment options for advanced or recurring cases. This research investigates the possibility of using Newcastle disease virus (NDV) along with Everolimus (EVE) and Beclin-1 (BEC) to improve immune reactions and decrease tumor development in an experimental model of HPV-related cervical cancer.

### Methods

A mouse model for cervical cancer was created by utilizing HPV16 E6/E7-expressing TC-1 cells in C57BL/6 mice. The mice underwent treatment with NDV, EVE, BEC, or various combinations of these therapies. Tumor progression was monitored, evaluated immune responses by measuring cytokine levels (including IL-4, IFN-γ, and IL-12), and investigated the presence of CD8+T cells within the tumors. Additionally, survival rates were monitored throughout the study.

### Results

The synergy of NDV, EVE, and BEC led to a remarkable decrease in tumor growth, achieving reductions of as much as 70% when compared to monotherapies. Additionally, our combination therapy elicited strong immune reactions, evidenced by increased concentrations of IL-4, IFN-γ, and IL-12, along with enhanced infiltration of CD8+T cells into the tumors. Mice that were subjected to this Triple therapy exhibited better survival rates than those in other treatment categories.

**Data availability statement:** All relevant data are within the paper and its Supporting Information files.

**Funding:** This project was co-funded by the Guilan University of Medical Sciences and the Pasteur Institute of Iran under project code **99022904**. The funding bodies provided financial support for research materials, animal studies, and personnel involved in the experimental work. The funding sources had no role in the design of the study, data collection, analysis, interpretation, or in writing the manuscript.

**Competing interests:** The authors have declared that no competing interests exist.

## Conclusions

Our findings highlight the potential to improve outcomes in cervical cancer associated with HPV through a multi-faceted approach incorporating NDV, Everolimus, and Beclin-1. This therapeutic strategy not only hinders tumor growth but also strengthens the immune system's ability to fight against cancer. These results prompt further exploration of this combination in clinical trials, with the goal of offering new treatment avenues for patients who have limited choices.

## Introduction

Chronic infection with high-risk types of human papillomavirus (HPV) particularly, HPV16 and HPV18 that have been associated with the majority of cervical cancer cases because of their ability to integrate into the host genome and drive oncogenic processes through the expression of the E6 and E7 oncoproteins, which deactivate tumor suppressor proteins like p53 and Rb, resulting in unregulated cell growth and malignant transformation [1–4]. Despite preventive vaccines and screening programs available for most, options for effective therapeutics in the case of advanced or recurring cervical cancer remain very limited, thus underlining the urgent need for novel treatment strategies [1].

One promising method of interest for cancer therapy is the approach involving oncolytic viruses [2]. OVs are engineered to selectively infect and destroy tumor cells while stimulating an anti-tumor immune response [3]. Recent studies have underlined the dual mechanism of OVs: direct oncolysis and induction of immunogenic cell death (ICD) with the release of damage-associated molecular patterns (DAMPs) and tumor-associated antigens (TAAs), which favor the activation of innate but also adaptive anti-tumor immunity [4,5]. In particular, Newcastle Disease Virus (NDV) is an RNA virus that has elicited much interest as an oncolytic agent because it undertakes most of the putative functions such as induction of apoptosis in tumor cells, enhancement of systemic anti-tumor immunity, and increased production of the ICD [6,7].

Beyond oncolytic viruses (OVs), there are various combinations of these agents with different treatment methods, including mTOR (mammalian target of rapamycin) inhibitors and autophagy modulators. Everolimus, which is an mTOR inhibitor, interferes with essential signaling pathways that play a role in cell proliferation and survival, thereby improving the effectiveness of standard therapies [8,9]. Autophagy is generally considered a process involving the degradation of cellular components and can be modulated either to promote or inhibit tumor progression. Beclin-1 represents one of the key regulators of autophagy that has been identified as a potential target in attempts to enhance the tumor sensitivity to treatment through the induction of autophagic cell death. [10–12]. Therefore, combining oncolytic virotherapy as an inducer of ICD with Everolimus and Beclin-1 presents a multi-interventional approach in attempting to tackle cervical cancers [13]. The combination of NDV, EVE, and Beclin-1 constitutes a multifaceted treatment approach designed to address tumor resistance and boost immune responses through increased ICD induction or

improved tumor-infiltrating CD8+T cells [8,14]. NDV causes oncolysis and activates the immune system, Everolimus hinders tumor cell proliferation by focusing on the mTOR pathway, and Beclin-1 regulates autophagy to enhance tumor cell death. This method provides a holistic solution to the drawbacks of current therapies for HPV-related cervical cancer by combining virotherapy, mTOR inhibition, and autophagy modulation.

Therefore, this study investigates the synergistic effects of NDV, Everolimus, and Beclin-1 in a mouse model of cervical cancer to evaluate their potential as an integrated therapeutic strategy. We propose that combining these therapies will lead to improved tumor growth suppression, strong immune activation, and beneficial modulation of the tumor microenvironment, offering a new treatment approach for cervical cancer.

## Materials and methods

### Cell lines and culture conditions

The TC-1 cells were grown in complete RPMI (Roswell Park Memorial Institute) 1640 medium [Gibco BRL, Gaithersburg, MD] containing 10% fetal bovine serum (FBS) [Gibco, Rockville, MD], 100 units/mL of penicillin, 100 μg/mL streptomycin and respectively, 0.4 mg/mL G418 [all from GIBCO, UK], 0.5 mM sodium pyruvate [Sigma Aldrich, Germany] and 2 mM L-glutamine [15]. The EL4 cell line represents a murine T-cell lymphoma and was maintained in RPMI 1640 containing 10% FBS; cells were cultured in a humidified incubator at 37°C with 5% CO2. Cells were passaged using 0.25% trypsin-EDTA and subcultured at a 1:3 dilution every 2–3 weeks.

### Virus preparation

The LaSota strain of Newcastle disease virus was prepared by the Razi Institute of Serum and Vaccine Research Center. For virus production, NDV was grown in the allantoic cavity of 9- to 11-day-old SPF embryonated chicken eggs. The collected allantoic fluid was stored at -80°C. Virus titers were determined using the EID50 method and specified the NDV titer concentration utilized for treatment ($10^8$ PFU/mL) in the experiment. For inactivation experiments, NDV was inactivated by UV treatment, and inactivation was verified by the absence of a cytopathic effect or plaque formation in Vero cells. The conversion factor used was EID50 ~ 0.7 PFU.

### Preparation of everolimus

Everolimus (Sigma Aldrich, Germany) was dissolved in a vehicle composed of 50% dimethyl sulfoxide (DMSO), 40% propylene glycol, and 10% absolute ethanol. To enhance solubility, 0.4 μL/mL Tween 20 was added. This solution was then mixed with 10% FBS before intraperitoneal administration to murine C57BL/6 mice.

### Beclin-1 plasmid preparation

Subsequently, the Beclin-1 gene was cloned into the pcDNA3 vector, InvivoGen, USA. Restriction endonuclease digestion with XhoI and BamHI verified the recombinant plasmid with further confirmation from DNA sequencing. Plasmid preparation was performed using EndoFree plasmid Giga Kit (Qiagen, Hilden, Germany).

### Treatment groups and administration

Mice were randomly divided into ten groups (n = 10 for each group):

1. PBS control: Administered 100 μL of intratumoral PBS.

2. NDV alone: Administered intratumoral NDV ($10^8$ PFU/100 μL).

3. Everolimus alone: Administered intraperitoneal Everolimus (5 mg/kg).

4. Beclin-1 alone: Administered intratumoral Beclin-1 plasmid (100 μL of 1 μg/μL).

5. NDV and Beclin-1: The pairing of NDV with Beclin-1.

6. NDV with Everolimus: A mix of NDV and Everolimus.

7. Everolimus and Beclin-1: The pairing of Everolimus with Beclin-1.

8. Triple therapy: Combination of NDV, Everolimus, and Beclin-1.

Treatment protocols were given on two occasions, with a one-week gap separating the doses. NDV and plasmids were injected directly into the tumor, whereas Everolimus was administered intraperitoneally to achieve systemic mTOR inhibition.

## Animal model

In this study, six- to eight-week-old female C57BL/6 mice were obtained from the Institute Pasteur of Iran and kept in a specific pathogen-free condition. Mice were adapted to environmental conditions such as food, water and 12-hour light/dark cycle for one week before starting our experiment. One week after the mice were tumorized, our designed treatments were injected into them (details below). After a while, half of the animals in each group were randomly selected to have their spleens removed after anesthesia with ketamine and tested for immunological factors (the animals were at least 9 weeks old and at most 12 weeks old when the spleens were removed). The other half of the animals were examined daily to follow up the tumor volume up to 6 weeks after vaccination with digital calipers by using the formula: tumor volume = (length × width²)/2. It should also be noted that all mice were monitored twice daily until the end of the experiment. Animals that either survived until the end point of the experiment or were moribund were sacrificed via asphyxiation by $CO_2$ or exsanguination under deep ketamine anesthesia according to ethical protocols. The following criteria were used to identify moribund animals: a) complications of injection such as a wound, bleeding, or infection; b) clinical or behavioral signs unresponsive to appropriate intervention such as labored breathing, significant inactivity, sunken eyes, hunched posture, matted fur, one or more unresolved skin ulcers, and abnormal vocalization when handled persisting for 1 day. During our study design, Animals were monitored to ensure that tumors did not ulcerate and did not impede animal ambulation according to the Guilan Medical University Animal Care guidelines for tumor burden. All procedures related to the animals were carried out according to protocols by the Guilan Medical University Animal Care and Use Committee (ethics number: IR.GUMS.REC.1399.540). At the end of this experiment after 8 weeks, as I mentioned above all surviving animals were euthanized by asphyxiation with $CO_2$.

## In vivo tumor treatment

Mice were randomly assigned to 10 groups (n = 10 per group). Tumors were induced by subcutaneous injection of $7 \times 10^5$ TC-1 cells into the right flank. Treatments commenced ten days post-injection. Mice received intraperitoneal injections of Everolimus (5 mg/kg) and intratumoral injections of Beclin-1 plasmid (100 μL) and NDV ($10^8$ PFU/100 μL). Combinations included NDV-BEC, NDV-EVE, BEC-EVE, and NDV-EVE-BEC (Triple therapy), administered twice at one-week intervals. Control groups received PBS (100 μL), vehicle, or pcDNA3. Tumor growth and mouse survival were monitored bi-weekly. Tumor size was calculated using electronic calipers.

## Lymphocyte proliferation assay

Cell-mediated immunity was assessed by culturing splenocytes from treated mice ($2 \times 10^5$ cells/well) in 96-well plates in RPMI-1640, supplemented with 10% FBS, 1% L-glutamine, 1% HEPES, and 0.1% penicillin/streptomycin. Subsequently, the cells were stimulated for 72 h with 1 μg/mL E7-specific H-2Db CTL epitope, PHA as a positive control, or culture

 

medium alone. Cell proliferation was quantified using a tetrazolium-based assay. A stimulation index was calculated to quantify lymphocyte activation.

$$\text{Stimulation index} = \frac{\text{OD values of stimulated cells (Cs)}-\text{relative cell numbers of unstimulated cells (Cu)}}{\text{relative OD values of unstimulated cells}} \qquad (1)$$

### In vitro cytotoxicity assay

For the assessment of CTL activity, cells from the spleen of treated mice were used as effector cells. The EL4 was pulsed with 1 µg/mL E7-specific H-2Db CTL epitope and co-cultivated with effector cells at a 50:1 ratio. The cytotoxicity was based on lactate dehydrogenase (LDH) release. Supernatants, after centrifugation, were analyzed for LDH levels.

### Cytokine ELISA

Seven days post-treatment, spleens from mice (n = 3) were isolated; mononuclear cells were cultured in 24-well plates with RPMI-1640, 10% FBS, 1% L-glutamine, 1% HEPES, 2.5 mM 2-mercaptoethanol, and E7-specific H-2Db CTL epitope (1 µg/mL).After 48 hours, supernatants were collected and analyzed for IL-4, IFN-γ, and IL-12 using commercially available ELISA kits (R&D Systems Inc., Minneapolis, MN, USA) according to the manufacturer's instructions. Results were read at 450 nm.

### Statistical analysis

All statistical analysis was performed using the SPSS 16.0 software through the one-way ANOVA technique. A value of *$P < 0.05$, **$P < 0.01$, ***$P < 0.001$, and **** $P < 0.0001$ were considered to demonstrate statistical significance.

## Results

### Effects of combination groups on lymphocyte proliferation

To evaluate the E7-specific lymphoproliferative response resulting from the treatments, we performed a lymphocyte proliferation assay. Mice treated with the Triple therapy exhibited the most significant E7-specific T-cell proliferation compared to all other groups ($P < 0.001$) (Fig 1). Among the double therapy groups, NDV-EVE induced the highest level of E7-specific T-cell responses ($P < 0.0001$), after that NDV-BEC and EVE-BEC showed the surpasses level of simulations compared monotherapy and control groups ($P < 0.0001$). Notably, the responses in the NDV and EVE groups were not significantly different from each other but were significantly higher than those in the control and BEC groups ($P < 0.0001$). The BEC group did not elicit a significant increase in T lymphocyte proliferation compared to controls ($P > 0.05$) (Fig 1). These results indicate that the Triple therapy significantly enhances E7-specific T-cell responses.

### Enhanced cytotoxicity of CTLs

To further investigate the anti-tumor mechanism of the combination therapies, an LDH release assay was used to assess the cytotoxic activity of E7-specific CTLs. The highest specific target lysis was observed at an Effector: Target (EL4) ratio of 100:1. Mice treated with Triple therapy showed significantly higher E7-specific CTL responses compared to all other groups ($P < 0.0001$) (Fig 2). Following the Triple therapy, the NDV-EVE, NDV-BEC, and NDV groups displayed the highest E7-specific lytic activity, significantly outperforming other groups ($P < 0.0001$). The EVE-BEC and EVE groups also demonstrated increased CTL activity compared to BEC and control groups ($P < 0.0001$), although they did not differ significantly from each other. Among the monotherapy groups, the NDV group had the highest antigen-specific cytolytic response compared to the BEC ($P < 0.0001$) and EVE ($P < 0.001$), respectively (Fig 2). A negligible antigen-specific cytolytic response

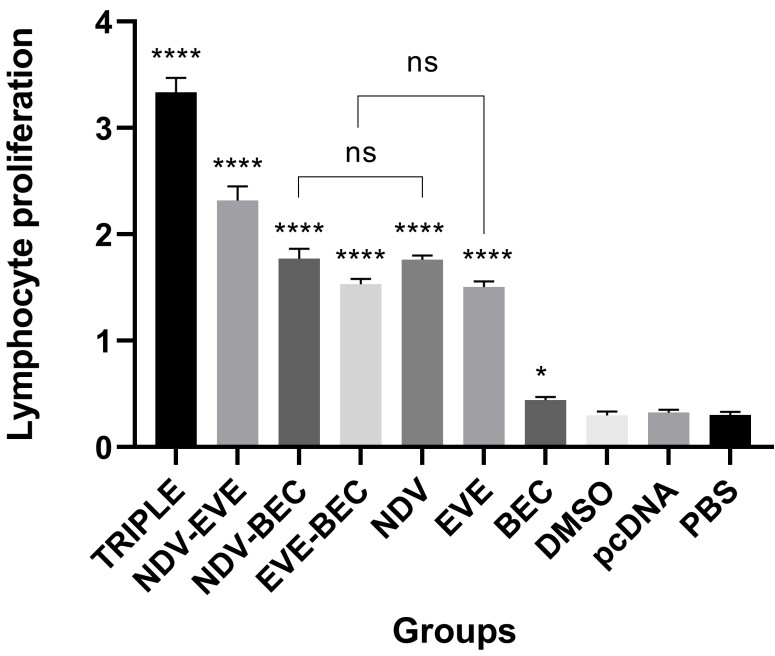

**Fig 1. Lymphocyte proliferation assay following combination therapy with NDV.** The Triple therapy group exhibited significant differences in lymphocyte proliferation compared to all other groups (**** P < 0.0001). Double therapy groups showed significant stimulation compared to monotherapies and control groups (**** P < 0.0001).

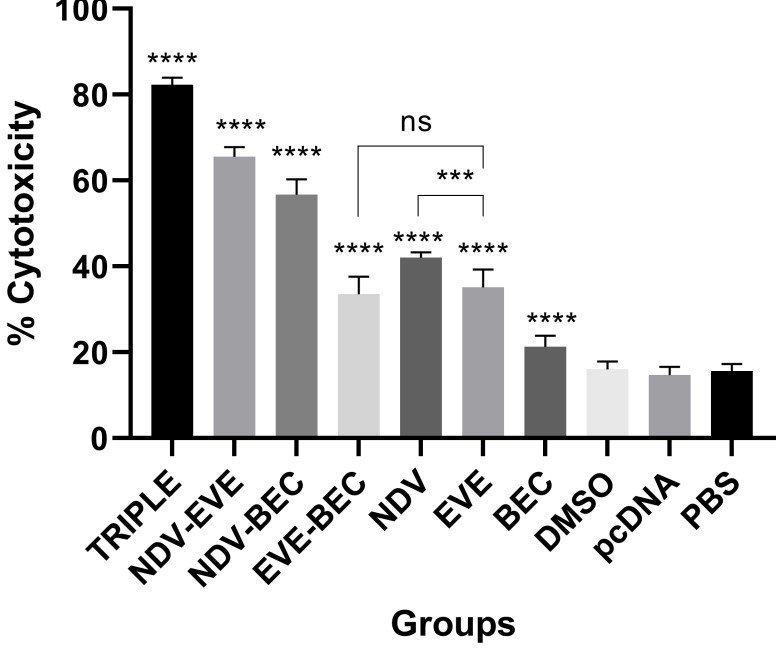

**Fig 2. CTL response following combination therapy with NDV.** The Triple therapy group significantly increased CTL responses compared to all other groups (**** P < 0.0001). Cytolytic analysis showed increased CD8 + activity in NDV-EVE, NDV-BEC, and NDV groups compared to monotherapy and control groups (**** P < 0.0001).

was detected in the control groups (pcDNA, DMSO, and PBS). These findings highlight that the combination therapies, particularly those involving NDV, enhance specific cytotoxic responses against TC-1 cells in the syngeneic model.

### Induction of anti-tumor cytokine secretion

To assess anti-tumor activity, we analyzed the secretion of Th1 (IFN-γ) and Th2 (IL-4) cytokines from restimulated splenocytes. The Triple therapy group induced the highest IFN-γ levels (P<0.0001) compared to all other groups ([Fig 3A]). NDV-EVE-treated mice also exhibited significantly higher IFN-γ levels than those treated with NDV-BEC, EVE-BEC, and

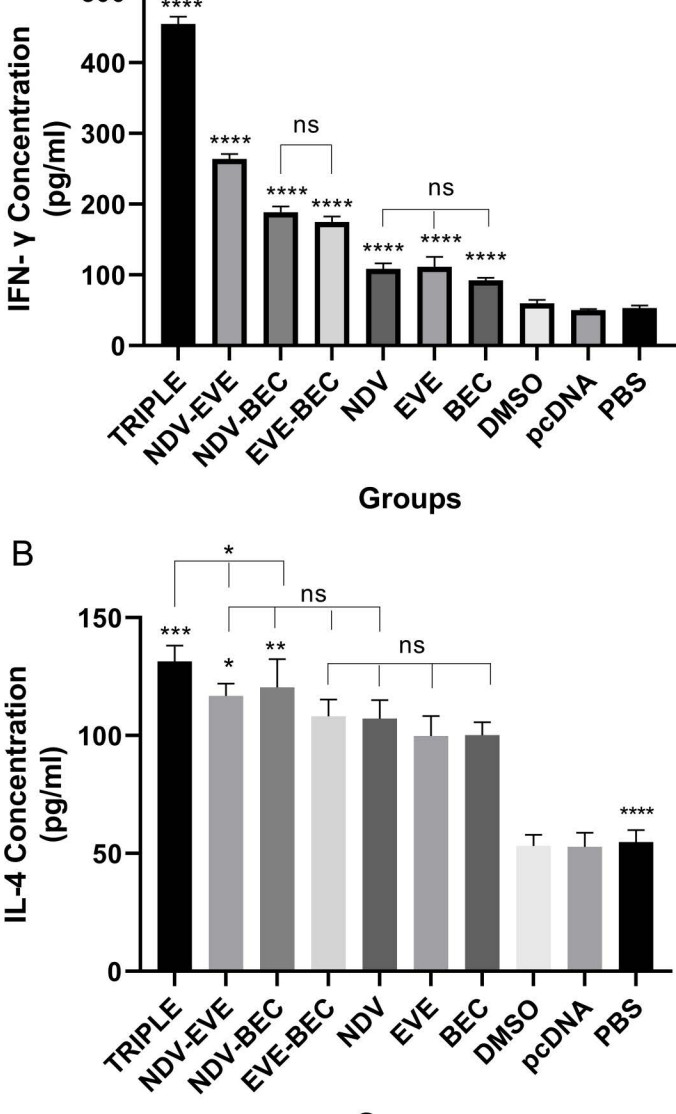

**Fig 3. Determination of the splenic IFN- γ, and IL-4 cytokine secretion. (A)** The Triple therapy group had the highest IFN-γ production compared to all other groups (**** P<0.0001). **(B)** IL-4 levels increased in all treatment groups compared to controls (**** P<0.0001), with the Triple therapy group showing the highest levels (***P<0.001). Each bar represents mean±SD from tumor samples of three mice per group.

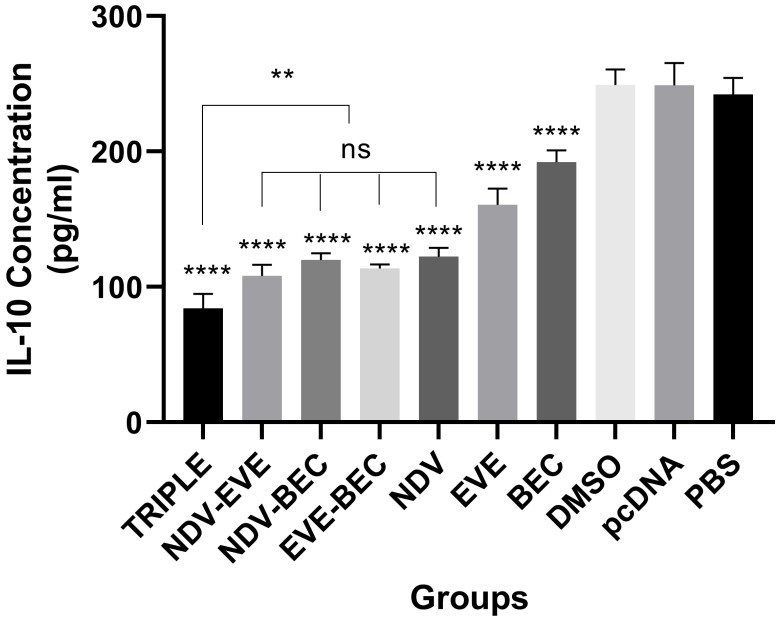

monotherapy groups (P<0.0001). NDV-BEC and EVE-BEC groups showed elevated IFN-γ levels compared to monotherapy and control groups (P<0.0001), but not significantly different from each other. It is worth noting that no significant difference in IFN-γ concentration was observed between the NDV, EVE, BEC with each other's, but the secretion of this cytokine in the three aforementioned monotherapy groups increased significantly compared to the control groups (P<0.0001) (Fig 3A).

Mice treated with the Triple therapy group showing the highest IL-4 concentration compared other treatment groups (P<0.001), although not significantly different from NDV-EVE and NDV-BEC (P<0.05) (Fig 3B). It should be noted that the IL-4 concentration in the NDV-EVE and NDV-BEC groups is also not significant compared to NDV and EVE-BEC, but they are slightly more significant compared to the other monotherapy groups (P<0.05). IL-4 levels were significantly higher in EVE, NDV, and BEC groups compared to controls (P<0.0001). IL-4 levels in the EVE, NDV, and BEC groups were not significantly different between each other, but we can observe noticeable difference compared to the control groups (P<0.0001), and also limited cytokine production was observed in control groups compared the other treatment groups (Fig 3B).

### Intratumoral cytokine levels

We measured intratumoral IL-10 levels to evaluate the impact of treatments on the tumor microenvironment. All treatment groups showed reduced IL-10 levels compared to controls (P<0.0001) (Fig 4). The Triple therapy group exhibited the lowest IL-10 levels, followed by NDV-EVE, NDV-BEC, EVE-BEC, and NDV groups (P<0.001) and EVE and BEC (P<0.0001), respectively. There were no significant differences in IL-10 levels between NDV-EVE, NDV-BEC, EVE-BEC, and NDV groups, however the graph illustrates the significant difference among these four groups compared with EVE and NDV (P<0.0001) (Fig 4). IL-10 is an immunosuppressive cytokine, and its reduction by combination therapy, particularly involving NDV, correlates with decreased tumor volume, as discussed further.

**Fig 4. IL-10 levels in tumor lysates post-treatment.** Significant reduction in IL-10 was observed in treatment groups compared to controls (**** P<0.0001). Each bar represents mean ± SD from tumor samples of three mice per group.

## Tumor growth inhibition

Tumor growth was monitored for 6 weeks following treatment. Mice in the Triple therapy and NDV-EVE groups exhibited significant tumor volume reduction compared to other groups (P<0.0001) (Fig 5). NDV-BEC and EVE-BEC groups also showed significant tumor control compared to monotherapy and control groups (**** P<0.0001), though they did not differ significantly from each other. At the end of the study, the maximum diameter of the tumor volume was measured in pcDNA (1288 mm³) and PBS (1256 mm³) groups. These results indicate that the combination therapies, especially those involving NDV, effectively modulate immune responses and enhance anti-tumor effects.

The appearance and weight details of the mice in all groups were analyzed from the first day to the end of the experiment. In the control groups, the mice were in normal conditions in terms of appearance and weight throughout the experiment. The mice in the treatment groups, especially the Triple therapy group, were also healthy in terms of appearance and weight characteristics, which indicates the absence of side effects of the treatment used in this experiment.

## Discussion

Cervical cancer remains a major challenge, especially in advanced or recurring instances where treatment alternatives are scarce. Our study investigates a novel combination therapy that capitalizes on the synergistic effects of Newcastle disease virus (NDV), Everolimus (EVE), and Beclin-1 (BEC) in treating human papillomavirus-associated cervical cancers in the mouse models. The approach now constitutes a major improvement in design over any prior strategies since it integrates oncolysis, autophagy modulation, and the inhibition of mTOR into the single formulation of the therapy. Recently, several studies have elaborated on the role of oncolytic viruses in cancer therapy, especially with regard to their potential to induce ICD, enhance antitumor immunity and that eliminate lesions and malignant tumors through cell-mediated immune responses against HPV-infected cells [16,17]. Extensive studies have been carried out with Everolimus regarding its role in the inhibition of the mTOR pathway, which plays an important role in cell growth and survival. Guglielmelli P et al. observed that Everolimus, as a single agent, can suppress tumor growth in various cancers by inhibiting mTOR signaling [18]. Autophagy has a very complex role in cancer [19], and

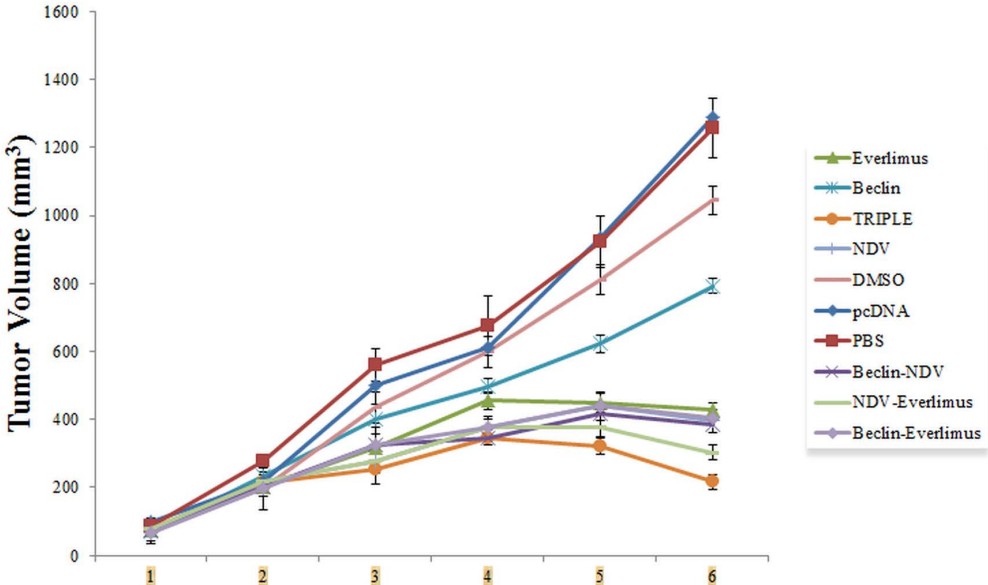

**Fig 5. Tumor volume measurements.** Significant reductions in tumor size were observed in the Triple therapy and NDV-EVE groups compared to controls (**** P<0.0001). Tumor growth inhibition was also significant in NDV-BEC and EVE-BEC groups (**** P<0.0001).

in our study, we utilized Beclin-1 to drive autophagy toward promoting tumor cell death rather than survival, thereby increasing the efficacy of our combination therapy. This approach is supported by the work of Pérez-Hernández M et al., who showed that enhancing autophagy can sensitize tumors to chemotherapy and reduce resistance [20]. The results indicated by Huangfu L et al., stating that in preclinical models, mTOR inhibitors combined with autophagy inducers significantly improved therapeutic outcomes [21]. Another study tested Everolimus in combination with an oncolytic adenovirus. While RAD001 seemed to interfere with the viral replication in vitro, strong anti-cancer effects were observed in vivo-presumably due to induction of autophagic cell death [22]. Our study illustrates that combining Everolimus with NDV and Beclin-1, which not only inhibits mTOR but also enhances autophagy, leading to increased tumor cell death. Oncolytic immunotherapy has been shown to release a wide range of damage-associated molecular patterns (DAMP) and tumor-associated antigens (TAA) from whole tumor cells through oncolytic virus replication that would be taken up and cross-presented to CD8+CTL T cells by activated dendritic cells, consequently leading to the activation of a tumor-targeting immune response [22]. Also, Kazeruni et al. showed that NDV increases the effectiveness of doxorubicin in cervical cancer, in part due to immune infiltration (CD8+T cell infiltration) and apoptosis in the tumor microenvironment [23]. Our results validate and broaden these observations, demonstrating that NDV works together with Everolimus and Beclin-1 to enhance. Further, the current study, OVs combined with immune checkpoint inhibitors and autophagy modulators markedly improved CD8+T-cell responses and tumor regression in solid tumors and pointed to the role of OVs in priming the tumor microenvironment for immune-mediated attack. In this regard, the induction of ICD by Triple combination therapy may be activating the immune cells including CTLs, enhancing the infiltration and activity of CD8+T cells and causing release of inflammatory responses within the tumor model. In order to find therapeutic effects caused by the Triple therapy in a human papillomavirus-associated tumor, we developed a syngeneic mouse model of papillomavirus-associated cancer using immunocompetent mice.

Recent research on mRNA-based HPV vaccines, such as those conducted by Lee et al. and Ramos da Silva et al., highlights the promise of mRNA technology in generating strong immune responses. Although mRNA vaccines provide immunity targeted at specific antigens, our NDV-based method provides the advantages of both direct tumor cell destruction and systemic immunogenic cell death [24].

In addition, our studies have revealed that the triple combination therapy is capable of offering improved antitumor efficacy in the TC-1 tumor mouse model, which is related to an increase in antigen-specific lymphocyte proliferation, CD8+cytotoxicity, and IFN-γ, IL-4, and IL-10 induction.

Th1 cells producing IFN-γ exhibit anti-cancer properties, while Th2 cells generating IL-10 and IL-4 support the humoral immune-response. Other research reveal that combination of NDV and chemotherapy drugs increase Th1 and also Th2 cytokines [23]. The other study believe that an effective HPV therapeutic vaccine should activate a potent cell-mediated immune response where CD4+T cells support CD8+T cells by secreting cytokines such as IL-4 and IFN-γ [25]. Dorostkar et al. emphasized the effectiveness of a STING activator paired with CpG-C adjuvants alongside an HPV16 E7 vaccine, which suppressed tumor proliferation and increased cytokine levels, such as IFN-γ and IL-4 [26]. Likewise, our Triple therapy greatly boosted these cytokines, indicating a strong Th1/Th2 immune response. In contrast to Dorostkar et al., our method combines virotherapy with autophagy and mTOR modulation, offering wider mechanisms to address tumor resistance. The decrease in IL-10 levels in the Triple therapy group indicates a suppression of immunosuppressive pathways, thereby boosting anti-tumor immunity. These results are consistent with Qi et al., who showed that multi-epitope vaccines aimed at HPV oncoproteins may lower immunosuppressive cytokines and encourage tumor shrinkage [27].

The detected immune activation, highlighted by elevated cytokines (IFN-γ and IL-4) and enhanced CD8+T cell infiltration, emphasizes this combination therapy's potential to alter the tumor microenvironment. This is vital for addressing immune evasion commonly utilized by HPV-associated tumors. The decrease in IL-10 levels in the Triple therapy group indicates a suppression of immunosuppressive pathways, thereby boosting anti-tumor immunity. These results are consistent with Qi et al., who showed that multi-epitope vaccines aimed at HPV oncoproteins may lower immunosuppressive cytokines and encourage tumor shrinkage. The results showed that immunization with the Triple therapy group elicited a

strong cell-mediate immunity as the highest rate of lymphocyte proliferation and IFN-γ and IL-4 levels were evaluated in this group, and the difference between this group (Triple therapy) and the other combination and monotherapy groups was statistically significant and recommended this kind of treatment is a reliable therapy which provoked a moderately effective immune response against TC-1-induced tumors and caused suppression of tumor growth. However, the Triple therapy group significantly prolonged survival that was potentially associated with upregulation of E7-specific CTL and Th1 responses when compared to other-treated mice. These data suggest that using combination therapy simultaneously had synergetic effect to elicit a potent cell-mediated immunity leading to inhibition of tumor growth.

Results also showed that the capability of the Triple combination therapy in inducing a robust antigen-specific cytolytic immune response causes a high antitumor activity against E7-expressing TC-1 tumor murine model, slowing tumor growth in tumor treatment experiments in vivo. As a result, the application of the Triple therapy has been preferred due to lower undesirable systemic toxicity compared to monotherapy. The potential of our findings is significant, especially given the robust in vivo validation. Unlike many recent studies that focus primarily on in vitro models, our research provides extensive in vivo evidence, showing that our combination therapy not only inhibits tumor growth but also prolongs survival and induces a durable immune response. This positions our approach as a strong candidate for clinical trials, particularly in patients with HPV-associated cervical cancer who have limited treatment options.

## Conclusion

To sum up, the NDV-EVE-BEC combination demonstrates notable benefits in decreasing tumor growth and boosting immune reactions in cervical cancer linked to HPV. These findings set the stage for more in-depth exploration of this combination approach, which could lead to improved treatment choices for cervical cancer patients with few options available. Future studies should investigate how to apply these discoveries in a clinical setting, especially when used alongside other forms of immunotherapy, in order to achieve the best results for patients.

## Supporting information

**S1 Data. MTT assay results.** Raw and analyzed cell viability data (absorbance values) for cervical cancer cells treated with oncolytic NDV, Everolimus, Beclin-1 targeted therapies, and combinations. Includes averages and standard deviations (SD) across replicates (n = 6).
(XLSX)

**S2 Data. LDH cytotoxicity assay results.** Lactate dehydrogenase (LDH) release data (absorbance units) quantifying cancer cell death post-treatment. Contains raw values, averages, and SDs for all experimental and control groups (n = 7 replicates).
(XLSX)

**S3 Data. IFN-γ cytokine measurements.** Interferon-gamma (IFN-γ) levels (pg/mL) across treatment groups, demonstrating immune activation by combination therapies. Includes raw values, averages, and SDs (n = 5 replicates).
(XLSX)

**S4 Data. IL-4 cytokine measurements.** IL-4 secretion levels (pg/mL) in tumor microenvironment samples post-treatment with NDV, Everolimus, Beclin-1 modulators, and control groups (DMSO, pcDNA, PBS). Data include technical replicates and SD calculations.
(XLSX)

**S5 Data. IL-10 cytokine measurements.** IL-10 immunosuppressive cytokine data (pg/mL) for all experimental groups, highlighting reductions in IL-10 with TRIPLE therapy (NDV + Everolimus + Beclin-1 modulation).
(XLSX)

**S6 Data. Tumor volume measurements.** Longitudinal tumor growth metrics (mm³) for xenograft models treated with monotherapies and combinations. Includes individual time points (500–5000 cells) and SDs for replicates (n = 5 mice/group).
(XLSX)

## Acknowledgments

The authors would like to acknowledge Guilan University of Medical Sciences and Pasteur Institute of Iran.

## Author contributions

**Conceptualization:** Amir Ghaemi.

**Funding acquisition:** Amir Ghaemi, Mohammad Shenagari.

**Investigation:** Hedieh Zargaran, Mehdi Samadi.

**Methodology:** Mehdi Samadi.

**Project administration:** Amir Ghaemi, Mohammad Shenagari.

**Supervision:** Amir Ghaemi, Mohammad Shenagari.

**Visualization:** Amir Ghaemi.

**Writing – original draft:** Hedieh Zargaran.

**Writing – review & editing:** Amir Ghaemi, Mohammad Shenagari.

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
