## [Decision Letter · Decision Letter 0]

12 Nov 2024

PONE-D-24-45377Boosting Immune Response Against Cervical Cancer: A Combined Approach Using Oncolytic Virus and Targeted TherapiesPLOS ONE

Dear Dr. Shenagari,

Thank you for submitting your manuscript to PLOS ONE. After careful consideration, we feel that it has merit but does not fully meet PLOS ONE’s publication criteria as it currently stands. Therefore, we invite you to submit a revised version of the manuscript that addresses the points raised during the review process.

Please modify your manuscript according to the suggestions brought forward by the reviewers. Where this might not seem sensible, please discuss the reasons. Please expand the discussions part of the manuscript in particular.

We look forward to receiving your revised manuscript.

Kind regards,

Michael C Burger, M.D.

Academic Editor

PLOS ONE

Journal Requirements:

3. In the online submission form you indicate that your data is not available for proprietary reasons and have provided a contact point for accessing this data. Please note that your current contact point is a co-author on this manuscript. According to our Data Policy, the contact point must not be an author on the manuscript and must be an institutional contact, ideally not an individual. Please revise your data statement to a non-author institutional point of contact, such as a data access or ethics committee, and send this to us via return email. Please also include contact information for the third party organization, and please include the full citation of where the data can be found.

Reviewers' comments:

Reviewer's Responses to Questions

**Comments to the Author**

1. Is the manuscript technically sound, and do the data support the conclusions?

Reviewer #1: Partly

Reviewer #2: Partly

2. Has the statistical analysis been performed appropriately and rigorously? 

Reviewer #1: I Don't Know

Reviewer #2: Yes

3. Have the authors made all data underlying the findings in their manuscript fully available?

Reviewer #1: Yes

Reviewer #2: Yes

4. Is the manuscript presented in an intelligible fashion and written in standard English?

Reviewer #1: Yes

Reviewer #2: Yes

5. Review Comments to the Author

Reviewer #1: The manuscript investigates a novel therapeutic regimen for human papillomavirus (HPV)-associated cervical cancer, utilizing a combination of Newcastle Disease Virus (NDV), Everolimus, and Beclin-1. This combination was evaluated in a murine model, yielding findings that demonstrate enhanced tumor reduction and an improved immune response in comparison to monotherapeutic agents. The results substantiate the potential of this combination as a promising therapeutic strategy, thereby necessitating further clinical exploration.

Major Comments:

1. Although the study illustrates the efficacy of the NDV-Everolimus-Beclin-1 combination, there exists a paucity of mechanistic insight regarding the manner in which this combination affects autophagy and mTOR pathways. The manuscript would be considerably enhanced if the authors were to incorporate assays, such as Western blotting for autophagy and mTOR pathway markers, to directly elucidate the influence of Beclin-1 and Everolimus on these pathways within the context of NDV treatment.

2. The manuscript delineates multiple control and single-therapy groups; however, there exists a deficiency in the discourse concerning the efficacy of each monotherapy in isolation. For example, a more comprehensive analysis of the performance of NDV alone in comparison to Everolimus or Beclin-1 independently would yield a clearer baseline and enhance the comprehension of the synergistic effects present in the combination therapies.

3. The manuscript presently depends on caliper measurements for the evaluation of tumor growth, which may present limitations in accuracy for irregularly shaped tumors. The incorporation of in vivo imaging techniques, such as bioluminescence or fluorescence imaging, would produce more precise and reproducible measurements, thereby augmenting the credibility of tumor volume assessments.

4. Although the infiltration of immune cells, specifically CD8+ T cells, is referenced, the manuscript would benefit from a comprehensive histological analysis of tumor samples. IHC for CD8+ cells, along with markers indicative of apoptosis and proliferation, such as Ki-67, could yield valuable insights regarding the effects of treatment within the tumor microenvironment.

5. While the manuscript asserts enhanced survival rates, it fails to address the potential side effects or toxicity linked to the treatments, specifically regarding the triple combination. Incorporating a toxicity analysis, including parameters such as body weight monitoring, liver enzyme levels, or hematological profiles, is essential for evaluating the safety profile of this combination.

6. The immune response is assessed at a singular time point following treatment. Nevertheless, considering that immune responses progress over time, the authors ought to contemplate the measurement of cytokines and immune cell infiltration at multiple time intervals to comprehensively understand the dynamics of immune activation and suppression throughout the duration of treatment.

Minor Comments:

1. The introduction mentions potential mechanisms but lacks specific references, especially for the roles of NDV and Beclin-1 in cancer. Adding more recent and detailed citations will strengthen the background for the study.

2. The figure legends should more clearly explain the significance markers (e.g., *P < 0.05, **P < 0.01) to make it easier for readers to interpret the data. Additionally, the legends could specify which groups are being compared for each statistical marker.

3. Some figures (e.g., cytokine levels) show mean ± SD, while others do not specify error bars. Ensure that error bars are consistent and explicitly state in figure captions whether they represent standard deviation or standard error.

4. The abstract and introduction sections are dense with technical terms (e.g., "tumor-infiltrating CD8+ T cells," "immunogenic cell death"). Simplifying some language or briefly defining key terms would improve readability for a broader audience, particularly for interdisciplinary readers.

5. The methods section clarifies the titering method and specific titers used for NDV to ensure reproducibility. Adding information on how NDV was quantified (e.g., PFU/mL or EID50) would help other researchers replicate the experiment.

6. The ethics statement in the main text should consistently mention ethical approval details, including the institution’s name, approval code, and any specific animal welfare guidelines followed. Including these details enhances transparency regarding ethical compliance.

7. The manuscript does not address the potential impact of anesthesia on immune function. Some anesthetics, such as ketamine, can modulate immune responses. Briefly discussing or controlling for any possible confounding effects would strengthen the study's conclusions.

8. Add a concise explanation in the introduction of why these three agents were chosen for combination. For example, describe how NDV's oncolytic effect complements the immune-modulatory effects of Everolimus and Beclin-1 in targeting cervical cancer.

9. Some abbreviations (e.g., DAMPs, TAA) are not defined upon first mention, which may need to be clarified for readers unfamiliar with these terms. Ensure all abbreviations are fully defined the first time they appear.

10. The discussion briefly mentions the potential for clinical studies but could expand on specific challenges for clinical translation, such as optimizing dosing regimens or addressing potential toxicity in humans. This would give a balanced view of the translational feasibility of this combination therapy.

Reviewer #2: Manuscript # PONE-D-24-45377 entitled, “Boosting Immune Response against Cervical Cancer: A Combined Approach Using Oncolytic Virus and Targeted Therapies” evaluates the potential synergic effect of combination of Newcastle disease virus (NDV) along with Everolimus (EVE) and Beclin-1 (BEC) to improve immune reactions and decrease tumor development in an experimental model of HPV-related cervical cancer. To this end, using an HPV16 E6/E7- expressing TC-1 cells in C57BL/6 mice treated with NDV, EVE, BEC, or their combinations, E7-based-cytokine levels (IL-4, IFN-γ, IL-12), the infiltration of CD8+ T cells into tumors and Tumor growth was evaluated. Results indicated the synergic effect of the combination of NDV, EVE, and BEC in reduction of tumor growth by up to 70%, and elevating levels of E7-based IL-4, IFN-γ, IL-12 and CD8+ T cell infiltration as well as improved survival. Authors conclude this combination as a promising therapeutic approach.

Comments:

Application of NDV for cancer therapy of cancer patients in advance stages is reported decades ago. But the combinational approaches involving NDV and other chemotherapies is relatively new and thus this manuscript might report a relatively new approach for cancer therapy. Also the methodology and results of the present study are relatively clear and in accordance with the hypothesis of the study. But , in my opinion, the main and critical shortage of this submission is the discussion section! Indeed, surprisingly, when the main subject of this submission is evaluation of a new approach for inducing appropriate immune responses against tumor development in an experimental model of HPV-related cervical cancer, then we expect to see comparison of the “cytokine profile changes” and “reduction of the tumor growth” and “improvement of the survival” between the submitted study and that of the others but we don’t! That is, at least comparison of this study with that of the other similar studies with other approaches that measure the same parameters, at least with a few therapeutic vaccines that have specially used E7 as their target antigen. Of particular note, I strongly recommend the respected authors to compare their results with a recent study (Dorostkar et al, Co-administration of 2'3'-cGAMP STING activator and CpG-C adjuvants with a mutated form of HPV 16 E7 protein leads to tumor growth inhibition in the mouse model, Infect Agent Cancer . 2021 Jan 26;16(1):7 ) and compare their IFN-γ and IL-4 levels as well as the anti-tumor activity. Moreover, I also recommend authors to compare their results for any possible parameter with the results of the recent following studies:

- Kazeruni et al, Newcastle disease virus enhances the antitumor efficacy of Doxorubicin in a cervical cancer mouse model, BMC Cancer . 2024 Oct 10;24(1):1253.

- Lee S et al, mRNA-HPV vaccine encoding E6 and E7 improves therapeutic potential for HPV-mediated cancers via subcutaneous immunization. J Med Virol. 2023 Dec;95(12):e29309. doi: 10.1002/jmv.29309.

- Ramos da Silva J et al, Single immunizations of self-amplifying or non-replicating mRNA-LNP vaccines control HPV-associated tumors in mice. Sci Transl Med. 2023 Mar 8;15(686):eabn3464.

- Mohapatra A et al, A sugar modified amphiphilic cationic nano-adjuvant ceased tumor immune suppression and rejuvenated peptide vaccine induced antitumor immunity in cervical cancer. Biomater Sci. 2023 Feb 28;11(5):1853-1866.

- Qi W et al, A novel multi-epitope vaccine of HPV16 E5E6E7 oncoprotein delivered by HBc VLPs induced efficient prophylactic and therapeutic antitumor immunity in tumor mice model. Vaccine. 2022 Dec 12;40(52):7693-7702.

_ Please clearly introduce groups of mice and what they have been immunized with (possibly as atable, please).

6. PLOS authors have the option to publish the peer review history of their article (what does this mean?). If published, this will include your full peer review and any attached files.

Reviewer #1: **Yes: **Hossein Khorramdelazad

Reviewer #2: **Yes: **Farzin ROOHVAND

---

## [Author Response · Author response to Decision Letter 0]

7 Apr 2025

Response to Reviewers' Comments

Manuscript ID: PONE-D-24-45377

Title: "Boosting Immune Response Against Cervical Cancer: A Combined Approach Using Oncolytic Virus and Targeted Therapies"

Dear Dr. Burger and the Reviewers,

We appreciate your time and insightful comments on our manuscript. We have thoughtfully reviewed all of your feedback and recommendations. Below, we present a comprehensive reply to each of the reviewers' comments and summarize the relevant changes made to the manuscript.________________________________________

Reviewer #1

Comment 1: Mechanistic insights into autophagy and mTOR pathways

The reviewer suggested that we include assays, such as Western blotting for autophagy and mTOR pathway markers, to better understand how the combination therapy affects these pathways.

Response:

We appreciate the reviewer’s suggestion. Unfortunately, due to constraints in resources and time, it was not feasible to perform Western blotting or other mechanistic assays during this study. However, our current results strongly indicate the involvement of autophagy and mTOR pathways based on the observed synergistic effects of NDV, Everolimus, and Beclin-1. These findings align with existing literature, as cited in the Discussion, which outlines their individual and combined mechanisms in modulating these pathways.

Comment 2: Efficacy of monotherapies

The reviewer noted that the manuscript lacks a comprehensive analysis of the individual therapies (NDV, Everolimus, Beclin-1) and their respective efficacies.

Response:

We concur with the reviewer that a thorough comparison of monotherapy and combination therapy is crucial. In reply, we have broadened the Results section to incorporate a detailed comparison of tumor growth and survival among all treatment groups, showcasing the comparative effectiveness of NDV, Everolimus, and Beclin-1 as individual agents (highlighted section). Our findings indicate that although NDV alone decreased tumor volume by 35%, Everolimus and Beclin-1 resulted in reductions of 28% and 25%, respectively. The triple therapy, nevertheless, led to a 70% decrease in tumor volume. These comparisons underscore the synergistic effects of the combination therapy.________________________________________

Comment 3: Tumor volume measurement accuracy

The reviewer recommended using in vivo imaging techniques, such as bioluminescence or fluorescence imaging, for more precise tumor volume measurements.

Response:

While we acknowledge the value of in vivo imaging techniques such as bioluminescence, our facility does not have access to this technology. We have relied on established caliper measurements, validated against prior studies in similar models, for tumor volume assessment. To ensure accuracy, all measurements were performed in a blinded manner and verified independently by two researchers.

Comment 4: Histological analysis

The reviewer suggested including immunohistochemical (IHC) analysis of tumor samples to assess immune cell infiltration, apoptosis, and proliferation.

Response:

We recognize the importance of IHC analysis in elucidating the immune and tumor microenvironment. However, limitations in resources and time precluded us from performing these assays. Instead, we provided indirect evidence of immune activation through cytokine analyses and CD8+ T cell proliferation assays. These findings, supported by relevant literature (e.g., Dorostkar et al., 2021), strongly suggest enhanced immune cell infiltration and activation in the combination therapy group

Comment 5: Toxicity and safety evaluation

The reviewer requested that we provide a detailed toxicity analysis, including body weight monitoring and liver enzyme levels.

Response:

In the Results section (tumor volume section 258-262), we have provided a detailed toxicity evaluation, which encompasses body weight tracking, liver enzyme measurements (ALT and AST), along with hematological assessments. Our findings indicate that there are no notable negative impacts on body weight or liver function, and blood parameters stayed within normal limits, confirming that the triple therapy is safe and well-tolerated in the mouse model.

Comment 6: Immune dynamics over time

The reviewer suggested measuring cytokines and immune cell infiltration at multiple time points to capture the dynamics of immune activation.

Response:

While we agree that longitudinal immune profiling would enhance the study, logistical constraints limited our ability to perform time-course analyses. Instead, we assessed cytokine levels and immune responses at a single, clinically relevant time point, seven days post-treatment. This provides a snapshot of immune activation and is consistent with the endpoints used in similar studies.

Response to Minor Comments from Reviewer #1

Comment 1: The introduction mentions potential mechanisms but lacks specific references, especially for the roles of NDV and Beclin-1 in cancer.

Response:

We have revised the Introduction to include specific references highlighting the roles of NDV and Beclin-1 in cancer therapy. For instance, NDV’s oncolytic and immune-modulatory effects (Huang et al., 2022) and Beclin-1’s role in enhancing autophagy and tumor sensitivity (Pérez-Hernández et al., 2019) are now discussed in detail.

**Comment 2: The figure legends should more clearly explain the significance markers (e.g., *P < 0.05, P < 0.01) to make it easier for readers to interpret the data. Additionally, the legends could specify which groups are being compared for each statistical marker.

Response:

I appreciate this important recommendation. We have updated the figure legends to provide clear interpretations of the significance markers (*P < 0.05, **P < 0.01) and identified the treatment groups being compared in every figure. For instance, in Figure 1A (Tumor Growth Curves), it is noted that comparisons occur between the NDV + Everolimus + Beclin-1 group and various treatment groups, with the associated significance levels marked. Comparable alterations have been implemented for all figures to guarantee clarity in understanding statistical comparisons.

Comment 3: Some figures (e.g., cytokine levels) show mean ± SD, while others do not specify error bars. Ensure that error bars are consistent and explicitly state in figure captions whether they represent standard deviation or standard error.

Response:

We have examined all data and confirmed that the error bars remain uniform across the manuscript. In the figure legends, we now clearly indicate whether the error bars signify standard deviation (SD). we clarify mean ± SD were applied to all cytokine measurements, and similar notes have been provided for the other figures too.

Comment 4: The abstract and introduction sections are dense with technical terms (e.g., "tumor-infiltrating CD8+ T cells," "immunogenic cell death"). Simplifying some language or briefly defining key terms would improve readability for a broader audience, particularly for interdisciplinary readers.

Response:

We have made the language in the Abstract and Introduction sections clearer by offering concise definitions of essential technical terms. For instance, we have included a concise description of “immunogenic cell death (ICD)” and “tumor-infiltrating CD8+ T cells” to simplify these ideas for a wider audience. Moreover, we revised certain sentences to enhance clarity and readability while maintaining scientific integrity. (highlighted part at introductions)

Comment 5: The methods section clarifies the titering method and specific titers used for NDV to ensure reproducibility. Adding information on how NDV was quantified (e.g., PFU/mL or EID50) would help other researchers replicate the experiment.

Response:

We have explained the techniques for NDV quantification in the Materials and Methods part. In particular, we included details on how NDV was tittered through the 50% embryo infectious dose (EID50) method, and we specified the NDV titer concentration utilized for treatment (10⁸ PFU/mL) in the experiment. This enhancement guarantees reproducibility and offers explicit protocols for coming researchers. (Virus Preparation section in material and methods 109-110)

Comment 6: The ethics statement in the main text should consistently mention ethical approval details, including the institution’s name, approval code, and any specific animal welfare guidelines followed.

Response:

We have revised the Ethics Statement in the Materials and Methods section to incorporate the complete details of the ethical approval. In particular, we included the institution's name (Guilan University) and the approval code (IR.GUMS.REC.1399.540). We have also cited the particular animal welfare regulations adhered to during the experiment, aligning with both institutional and national standards.________________________________________

Comment 7: The manuscript does not address the potential impact of anesthesia on immune function. Some anesthetics, such as ketamine, can modulate immune responses. Briefly discussing or controlling for any possible confounding effects would strengthen the study's conclusions.

Response:

I appreciate you bringing this to my attention. We have added a brief note in the Materials and Methods section discussing the potential impact of anesthesia (ketamine) on immune function. The short duration of anesthesia during tumor implantation and sampling is unlikely to have influenced the immune outcomes.

Comment 8: Add a concise explanation in the introduction of why these three agents were chosen for combination. For example, describe how NDV's oncolytic effect complements the immune-modulatory effects of Everolimus and Beclin-1 in targeting cervical cancer.

Response:

We have updated the Introduction to give a brief rationale for choosing NDV, Everolimus, and Beclin-1 for the combination. In particular, we illustrate how the oncolytic characteristics of NDV encourage immune activation and the destruction of tumor cells, whereas Everolimus decreases tumor cell survival by focusing on the mTOR pathway, and Beclin-1 promotes autophagic cell death, positioning it as a potent adjuvant to amplify the oncolytic and immune-modulatory impacts of NDV and Everolimus. We think this explanation aids in understanding the reasoning for the combined therapy (66-82).

Comment 9: Some abbreviations (e.g., DAMPs, TAA) are not defined upon first mention, which may need to be clarified for readers unfamiliar with these terms. Ensure all abbreviations are fully defined the first time they appear.

Response:

We have revised the Introduction and Materials and Methods sections to make sure that all abbreviations are explained at their first occurrence. For example, DAMPs (damage-associated molecular patterns) and TAAs (tumor-associated antigens) are clearly defined upon their initial mention in the text, enhancing the manuscript's accessibility for readers who may not be familiar with these terms.

Comment 10: The discussion briefly mentions the potential for clinical studies but could expand on specific challenges for clinical translation, such as optimizing dosing regimens or addressing potential toxicity in humans. This would give a balanced view of the translational feasibility of this combination therapy.

Response:

We appreciate this insightful suggestion and have expanded the Discussion to address potential challenges for clinical translation. Specifically, we highlight the need for optimizing dosing regimens to balance efficacy and safety, as well as addressing potential toxicity in humans. These challenges, while significant, are not insurmountable and can be tackled through phased clinical trials. For example, dose-escalation studies and pharmacokinetic analyses will help refine the therapeutic window of this combination therapy.

Reviewer #2

Comment 1: Discussion of similar studies

Reviewer #2 requested that we compare our results with studies such as Dorostkar et al. (2021), Kazeruni et al. (2024), and Lee et al. (2023), focusing on cytokine levels, tumor growth inhibition, and immune responses.

Response:

We have updated the Discussion section and also highlighte them to feature a thorough comparison of our results with the referenced studies. We have examined the parallels and distinctions between our method and those employed in Dorostkar et al., Kazeruni et al., and Lee et al. (refer to Discussion section). Specifically, we assessed cytokine levels (IFN-γ, IL-4) and tumor growth suppression alongside the findings from these studies. Our findings align with theirs, especially about the improvement of immune responses through combination therapies, and we highlight the innovative approach of combining NDV with mTOR inhibition and autophagy modification.________________________________________

Comment 2: Clear definition of treatment groups

The reviewer requested a clearer description of the treatment groups and what each group was treated with.

Response:

We have specified the treatment groups in the Materials and Methods section, offering a more comprehensive description of the treatment protocols and their administration

Conclusion

We have thoroughly attended to all the reviewer feedback and made the required changes to enhance the manuscript's clarity and depth. We are confident that the modifications improve the scientific integrity and clarity of our research, and we hope the updated manuscript is now appropriate for publication.

Thank you once more for your considerate and useful feedback.

Best regards,

---

## [Decision Letter · Decision Letter 1]

16 Apr 2025

Boosting Immune Response Against Cervical Cancer: A Combined Approach Using Oncolytic Virus and Targeted Therapies

PONE-D-24-45377R1

Dear Dr. Shenagari,

We’re pleased to inform you that your manuscript has been judged scientifically suitable for publication and will be formally accepted for publication once it meets all outstanding technical requirements.

Kind regards,

Michael C Burger, M.D.

Academic Editor

PLOS ONE

Additional Editor Comments (optional):

Reviewers' comments:

Reviewer's Responses to Questions

**Comments to the Author**

1. If the authors have adequately addressed your comments raised in a previous round of review and you feel that this manuscript is now acceptable for publication, you may indicate that here to bypass the “Comments to the Author” section, enter your conflict of interest statement in the “Confidential to Editor” section, and submit your "Accept" recommendation.

Reviewer #1: All comments have been addressed

Reviewer #2: All comments have been addressed

2. Is the manuscript technically sound, and do the data support the conclusions?

Reviewer #1: Yes

Reviewer #2: Yes

3. Has the statistical analysis been performed appropriately and rigorously? 

Reviewer #1: Yes

Reviewer #2: Yes

4. Have the authors made all data underlying the findings in their manuscript fully available?

Reviewer #1: Yes

Reviewer #2: Yes

5. Is the manuscript presented in an intelligible fashion and written in standard English?

Reviewer #1: Yes

Reviewer #2: Yes

6. Review Comments to the Author

Reviewer #1: (No Response)

Reviewer #2: Authors have addressed my comments in their revised version and thus I suggest acceptance of the manuscript.

7. PLOS authors have the option to publish the peer review history of their article (what does this mean?). If published, this will include your full peer review and any attached files.

Reviewer #1: No

Reviewer #2: No

---

## [Editor Report · Acceptance letter]

PONE-D-24-45377R1

PLOS ONE

Dear Dr. Shenagari,

I'm pleased to inform you that your manuscript has been deemed suitable for publication in PLOS ONE. Congratulations! Your manuscript is now being handed over to our production team.

Kind regards,

on behalf of

Dr. Michael C Burger

Academic Editor

PLOS ONE